# Heart Rate Variability in Relation to Cardiovascular Autonomic Neuropathy Among Patients at an Urban Hospital in Kazakhstan

**DOI:** 10.3390/ijerph21121653

**Published:** 2024-12-11

**Authors:** Nazira Bekenova, Tamara Vochshenkova, Alisher Aitkaliyev, Balkenzhe Imankulova, Zhanatgul Turgumbayeva, Balzhan Kassiyeva, Valeriy Benberin

**Affiliations:** 1Medical Centre Hospital of President’s Affairs Administration of the Republic of Kazakhstan, Mangilik El 80, Astana 010000, Kazakhstan; nazira.bekenova@mail.ru (N.B.); aitkaliyev1998@gmail.com (A.A.); kassiyevabs@gmail.com (B.K.); valery-benberin@mail.ru (V.B.); 2University Medical Center Corporate Fund, Kerey and Zhanibek Khans St 5/1, Astana 010000, Kazakhstan; 3Institute of Innovative and Preventive Medicine, Alikhan Bokeikhan Street, Building 1, Astana 010000, Kazakhstan

**Keywords:** cardiovascular autonomic neuropathy, cardiovascular diseases, heart rate variability, autonomic nervous system

## Abstract

In clinical practice, heart rate variability (HRV) has not been considered an indicator for the preventive assessment of cardiovascular autonomic neuropathy (CAN). The paper studies HRV in a large, randomly selected group. A cross-sectional study included a representative sample of 5707 Kazakhs aged 20 years and older from a total population of 25,454 attached to an urban clinic in the capital of Kazakhstan. The sample was drawn from individuals who visited the clinic for a preventive examination. CAN diagnosis was confirmed using data from questionnaires, electronic medical records, HRV, and heart rate measurements. Mean values of the standard deviation of normal sinus RR intervals (SDNN) and the root mean square of successive RR interval differences (RMSSDs) from a 24 h electrocardiogram recording were assessed. CAN was identified in 17.19% of the study participants, with a ratio of the subclinical to clinical phase of 1:0.24. Diabetes mellitus was present in 30.99% of patients with CAN. The prevalence of CAN varied by sex and age, aligning with the prevalence trajectory of diabetes. It was concluded that the SDNN and RMSSD parameters in electrocardiographic studies can be used for preventive measures in the context of limited healthcare resources.

## 1. Introduction

Cardiovascular autonomic neuropathy (CAN) is characterized by dysfunction of the autonomic innervation of heart activity and vascular tone. Its development is associated with the widespread neuronal degeneration of small nerve fibers in both sympathetic and parasympathetic pathways, ultimately leading to neuronal death. The CAN subcommittee of the Toronto Consensus Group on Diabetic Neuropathy defines CAN as “a disorder of cardiovascular autonomic control in patients with established diabetes after other causes have been excluded” [1].

However, the pathophysiology of diabetic neuropathies involves many factors, and diabetic small-fiber polyneuropathy may precede *diabetes mellitus* (DM). Multifactorial interventions could potentially slow down the development of CAN [2,3,4].

CAN is an independent predictor of both all-cause mortality and cardiovascular mortality, with a projected mortality rate of up to 50% within the next 5 years in cases where clinical symptoms are present [5]. The association between CAN and impaired autonomic control of the cardiovascular system is a major cause of hidden cardiovascular events in patients without overt heart disease [6]. Although there is currently no compelling evidence for a cure for CAN, it is known that CAN development can be managed through a comprehensive reduction in cardiovascular risk factors. In this context, the early detection of autonomic dysfunction could be crucial for managing cardiovascular risk and could expand scientific efforts to find effective targeted interventions [5,7].

The causal relationship between heart rate variability (HRV) and cardiovascular diseases (CVDs) remains unclear despite the growing number of studies [8]. HRV is a quantitative method used to assess the balance between the sympathetic and parasympathetic nervous systems. It reflects fluctuations in autonomic signals to the heart, meaning that lower and higher HRV values can indicate abnormal conditions [9]. High HRV, as a surrogate marker for excessive autonomic fluctuations in patients with hypertension, may predict the development of atrial fibrillation independently of known cardiovascular risk factors [10]. Conversely, reduced HRV is associated with a higher risk of stroke, cardiovascular diseases, all-cause mortality, and mental disorders [11,12].

There are epidemiological data on the prevalence of CAN in diabetes, but there are significantly fewer data on the prevalence of HRV changes and their relationship to CAN. Such data could form the basis for planning more effective strategies for managing the early stages of CAN. This study aimed to investigate HRV changes in a large randomly selected patient group from real-world clinical practice.

## 2. Methods

### 2.1. Study Design and Patients’ Selection

This study was conducted at an urban clinic in Astana with an attached population of 25,454 people aged 20 years and older. The attached population of the clinic shares similar occupational backgrounds (government employees), lifestyle, and environmental factors, which may reduce the influence of external factors on the study results. The cohort was randomly selected from individuals who visited the clinic for preventive examinations from January 2023 to December 2023. Stratification included three age groups (20–39 years, 40–59 years, 60 years and older) with clustering by sex (men and women) within each age group. The representative sample size was calculated using Epi Info statistical software.

Inclusion Criteria: Age 20 years and older, of Kazakh ethnicity in the third generation.

Exclusion Criteria: Atrial fibrillation, complete AV block, implanted antiarrhythmic devices, cancer, pregnancy.

The participants self-reported their ethnicity in the questionnaire item “I identify myself, my biological parents, and my biological grandparents on both maternal and paternal lines as ‘Kazakh’”. The questionnaire also included questions to determine the presence or absence of orthostatic hypotension.

Data from individual electronic medical records were used to identify diagnoses of diabetes mellitus (E10–E14) and cardiovascular diseases (I10–I25), as well as autonomic diabetic neuropathy (G63.2; E10–E14+ with a general fourth character of 4) according to the International Classification of Diseases and Related Health Problems (ICD-10). Demographic data from the 5707 study participants were compared with HRV parameters, resting heart rate, and prevalence data for diabetes mellitus, cardiovascular diseases, and CAN. The disease (morbidity) prevalence was calculated based on the number of cases per 10,000 study participants in 2023.

### 2.2. Twenty-Four Hour Holter Electrocardiogram Registration and HRV Analysis

To ensure reproducibility, standardized procedures were used for HRV measurements. HRV was calculated based on 24 H Holter electrocardiogram (ECG) monitoring of the participants during their daily activities. The parameters of HRV were presented as the mean of the standard deviation of NN intervals (SDNN) for all 5 min segments of the entire recording and the Root Mean Square of Successive Differences (RMSSDs) [13]. SDNN measures the variability in time between consecutive heartbeats (NN intervals). It reflects overall heart rate variability and is calculated as the standard deviation of all NN intervals over a specified period. A higher SDNN indicates greater variability and a healthier autonomic nervous system. RMSSDs measure the variability between successive heartbeats. Specifically, it is the square root of the average of the squares of the differences between successive NN intervals. RMSSDs are particularly sensitive to changes in parasympathetic activity and are often used to assess autonomic function. Usually, higher RMSSD values indicate better parasympathetic regulation and overall heart health [13]. These time-domain parameters reflect long-term HRV: low values indicate a sympathovagal imbalance, leading to sympathetic hyperactivity, contributing to CAN. Since the purpose of this study was not to investigate risk factor associations, we did not focus on nocturnal values.

The 24 H ambulatory ECG monitoring was obtained using a 3-channel digital recorder (Medilog DARWIN, Switzerland) following the recommendations of the European Society of Cardiology and the North American Society of Pacing and Electrophysiology, as well as the Russian Federation [13,14]. The ECG recordings were processed automatically and manually, with any inappropriate computer-generated labels modified. All 24 H RR interval data from one heartbeat to the next were divided into 5 min segments, resulting in 288 segments of 5 min RR data. HRV was assessed using time-domain analysis.

Currently, HRV parameters do not have standardized values for diagnosing CAN. For accurate characterization of the HRV parameters, taking into account the patients’ age, we used the data from Umetani K et al. [15] and formed groups with normal, low, and high SDNN, RMSSDs, and heart rate (HR) values (Table 1). This approach allowed for ensuring the objectivity of the relationship between high or low heart rate and HRV indicators with the risk of cardiovascular morbidity and mortality (Table 1).

The signs of the subclinical phase of the autonomic nervous system (ANS) were identified as the detection of below the lower values of one of the indicators, either SDNN or RMSSDs, with an average HR value (Table 1). Signs of the clinical phase of CAN were identified as having at least one low HRV parameter with a high HR and the presence of pre-syncopal or syncopal episodes upon sudden changes in body position or physical exertion, unrelated to other causes. Our choice was based on previously established proposals for staging CAN depending on the level of damage to the ANS [16] and the severity of cardiovascular reflexes [1].

### 2.3. Statistical Analysis

Statistical analysis was conducted using IBM SPSS Statistics 26 for Windows (SPSS Inc., Chicago, IL, USA). Quantitative data were presented as means and standard deviations and used as continuous variables. Intergroup differences in age, SDNN, RMSSDs, and HR were assessed using Student’s t-test. Differences were considered statistically significant at *p* < 0.05.

### 2.4. Ethics

This study complies with the Declaration of Helsinki 1964 and Good Clinical Practice guidelines. This study was conducted in adherence to the ethical guidelines and received approval from the Hospital’s Local Commission on Bioethics, with permission note No. 5 issued on 27 September 2017. All methods were conducted following the recommendations. All participants were informed and provided written consent to participate in this study and for the use of their data for research and educational purposes, with confidentiality of personal information assured.

## 3. Results

### 3.1. Comparison of Clinical and Demographic Indicators of Patients with CAN

The study cohort included 5707 participants, with the sample selected to represent the attached population of the urban clinic (Table 2). CAN was identified in 981 participants, of whom 304 had a previously established diagnosis of DM. The prevalence of CAN is intermediate between CVD and previously diagnosed DM. CVD, type 2 DM, and CAN are more prevalent among men across all age categories, with their prevalence increasing with age and gender differences diminishing over time (Table 2).

In the overall group of patients without CAN, the SDNN and RMSSD values were statistically significantly lower in women compared to men. Heart rate was significantly higher in females compared to males. In the age group over 60 years, the mean RMSSD values for women were slightly lower but did not reach statistical significance (Table 3).

Regarding patients with CAN, there were no statistically significant differences in SDNN and RMSSD values or heart rate based on sex. In the age group of 20–39 years, the mean SDNN was lower in men. For other parameters, there were no significant differences based on sex among patients with CAN. Among those older than 60 years, although the mean values of SDNN, RMSSDs, and heart rate were lower in women compared to men, these differences were not statistically significant (Table 3).

### 3.2. Phenotypic Analysis of Study Participants

#### Phenotypic Groups

1.Normal HRV Phenotype

The group with normal SDNN and RMSSD values (within 95% CI) is the largest, comprising 64.69% of all study participants (Figure 1). The prevalence of this phenotype decreased 2.15-fold in the age group “60 years and older.” Women predominate in this group, and average values converge with age. There are significant differences (*p* < 0.05) in SDNN, RMSSDs, and HR between men and women. Men have higher normal HRV values, which decrease more rapidly with age compared to women. HR was higher in women across all age groups, with a more intense increase with age compared to men. Gender and age characteristics in this group primarily reflect the physiological autonomic activity of the cardiovascular system (CVS) in men and women (reduction in ANS activity with age; parasympathetic dominance in men). No cases of CAN or DM were found in this group.

2.Low HRV Phenotype

The group with low SDNN and RMSSD values (below 95% CI) is the smallest, comprising 4.08% of all study participants (Figure 1). The prevalence of this phenotype increased 1.84-fold in the age group “60 years and older.” Participants in this group are older on average compared to other age groups. Men predominate, and average values converge with age. There are no significant differences in SDNN, RMSSDs, and HR between men and women. All participants in this group have CAN, with DM present in 75.11% of participants. The subclinical phase of CAN predominates in the age group “20–39 years” with a ratio of 1:0.77; in subsequent age groups, the clinical phase takes precedence.

3.High HRV Phenotype

The group with high SDNN and RMSSD values (above 95% CI) comprises 5.19% of all study participants (Figure 1). The prevalence of this phenotype decreased 2.19-fold in the age group “60 years and older”. Men predominate, and average SDNN and RMSSD values are higher compared to women. Gender and age characteristics align with the physiological features of parasympathetic dominance in the ANS in men. No cases of CAN or DM were found in this group.

4.Transitional HRV Phenotype

The group with one HRV parameter (either SDNN or RMSSDs) above or below the 95% CI comprises 26.04% of all study participants (Figure 1). The prevalence of this phenotype increased 3.16-fold in the age group “60 years and older” mainly due to an increase in participants with low RMSSDs. Among cases with low HRV values across all age groups, the subclinical phase of CAN predominates with a ratio of 1:0.14; DM is present in 17.25% of participants with CAN. Gender and age characteristics in this group mainly reflect a mixed level of autonomic activity in the CVS.

## 4. Discussion

This study measured HRV parameters (SDNN, RMSSDs, and HR) in a large, randomly selected cohort of Kazakhs (Table 3). The results confirmed that age and gender influenced HRV in participants without CAN. Those in the “normal HRV” group, reflecting physiological autonomic activity, showed decreased sympathetic activity and parasympathetic dominance with age, a pattern also observed in the “high HRV” group.

The established links between low HRV and CAN highlight HRV as an important indicator of heart autonomic health in clinical practice, alongside standard cardiovascular reflex tests [3,15,17]. Despite over 5000 PubMed mentions in the past 3 years, traditional HRV methods have not been widely adopted for diagnosing or assessing CAN risk. This is partly due to various sources of HR fluctuations, which affect the reliability of HRV interpretation [18]. Additionally, HRV primarily reflects current metabolic status, limiting its long-term prognostic value. To address this, we used the 95% CI for 24 H SDNN and RMSSD measurements from Umetani et al. [15]. A 24 H HRV measurement does not significantly improve the detection of mental and physical health issues [19]. A 10 s ECG can also capture vagally mediated HRV with similar accuracy. Many wearable devices, despite technical limitations, can measure HRV, offering additional opportunities for patient-centered health management [19,20]. Following the goal of our study and considering existing research, we limited our analysis to SDNN and RMSSD indices without evaluating additional parameters [5,21].

Based on scientific data on autonomic control changes in CAN, we divided the condition into subclinical and clinical phases. Among 981 identified CAN cases, the subclinical phase predominated with a ratio of 1:0.24. DM was present in 15.84% of subclinical cases, while 93.23% of clinical cases had DM.

The ANS involvement in pathology progresses gradually, starting with structural and electrical remodeling. The vagus nerve, the longest parasympathetic nerve, is typically the first affected. Its dysfunction leads to chronic adrenergic stimulation, contributing to early electrical remodeling in CAN and parasympathetic neuropathy. Parasympathetic dysfunction is a strong predictor of poor CVD prognosis, triggering chronic inflammation and elevated heart rate. At this stage, changes in autonomic control are asymptomatic and detectable only through HRV. As sympathetic nervous system degeneration advances, structural changes occur, including increased sympathetic nerve density in the atria. HR and HRV become less responsive to sleep, activity, or stress, setting the stage for the clinical phase of CAN. Early clinical signs include tachycardia and orthostatic hypotension resulting from damage to sympathetic vasomotor fibers, particularly those affecting the visceral blood vessels [22,23,24].

Körei AE et al. found parasympathetic neuropathy signs in 56.8% of metabolic syndrome patients and 32.1% of prediabetic patients, highlighting the role of insulin in peripheral nerve function [3]. However, multifactorial factors—such as genetic, epigenetic, autoimmune, and toxic influences—can damage nerves independently or worsen hyperglycemia’s effects [25]. The hypothesis that genetic factors link CAN with low HRV underscores its relevance for early CVD development [26,27]. Elaine M. Urbina et al. suggest that CAN does not cause structural heart changes but emerges due to increased arterial stiffness, unrelated to diabetes type. In prediabetes, early small-fiber neuropathy affects around 40% of patients and may predict microvascular complications, including diabetic neuropathy [28]. In prediabetes, early small-fiber neuropathy affects about 40% of patients, and emerging autonomic dysfunction may predict microvascular complications, including diabetic neuropathy [29].

Diabetes damages the ANS more aggressively and is often linked to the clinical phase of CAN. In the “Low HRV” phenotypic group, all participants had CAN, with 75.11% also having diabetes. In the “20–39 years” age group, the subclinical phase predominated (ratio 1:0.77). However, after age 39, the clinical phase of CAN became more common, coinciding with the rising prevalence of diabetes.

Among the 5707 study participants, CVD prevalence was significantly higher than autonomic neuropathy (AN) and diabetes (DM). In younger individuals, CVD prevalence closely mirrored DM and AN patterns, but in those 60 and older, the sharp rise in CVD was likely influenced by additional factors. Dysglycemia and hyperglycemia play a direct role in CAN onset and progression. Autonomic dysfunction in the heart can develop due to poorly controlled glucose levels before changes appear in long-term markers like HbA1c and C-peptide, with HRV reflecting these early changes [30]. In our study, a decrease in SDNN or RMNN, along with a normal resting heart rate and absence of orthostatic hypotension, indicated parasympathetic dysfunction. This is promising, as research suggests that parasympathetic denervation may be reversible through multifactorial interventions in metabolic syndrome and prediabetes [24,31].

This study found AN prevalence to be 17.19% in random patients and 100% in those with DM, aligning with previous research showing that AN affects at least 20% of random patients and up to 65% of those with DM [24]. These results highlight the widespread impact of AN on heart autonomic control in the Kazakh population, consistent with the link between low HRV and DM [5,32]. As heart health studies increasingly use advanced neural networks, HRV indicators are likely to gain more attention [33]. A low HRV indicator without clinical signs of AN may signal the subclinical phase of AN, supporting early intervention for metabolic changes. Thus, routine HRV monitoring is crucial, beyond simply confirming AN as a DM complication [34].

We acknowledge this study’s limitations, including the inability to assess causal relationships with other factors. However, this study’s single-center design involved a population with similar activity levels, lifestyle, and environmental factors, which helps minimize external influences. The random sample included both healthy and sick individuals, and the cross-sectional approach provided an objective assessment of AN prevalence. Although only two HRV indicators were used, established threshold values helped address this limitation. Age-specific HRV values also allowed for a clear differentiation between low HRV due to disease and normal aging, meaning that they are nearing being well suited for clinical practice.

## 5. Conclusions

The key results of our study are the expanded opportunities for early diagnosis and treatment of AN at the level of routine clinical practice, even before precise diagnostic criteria are established.

In a population-based cross-sectional study of HRV conducted in Kazakhstan, phenotyping based on HRV parameters identified AN in 17.19% of participants. The ratio of subclinical to clinical phases of the disease was 1:0.24. The prevalence of AN varied by sex and age, following trends seen in DM prevalence. Therefore, using phenotyping results based on at least one temporal HRV parameter can improve the early detection of AN in clinical practice. As more heart health studies employ advanced neural network techniques, the significance of HRV indicators is expected to be re-evaluated.

## Figures and Tables

**Figure 1 ijerph-21-01653-f001:**
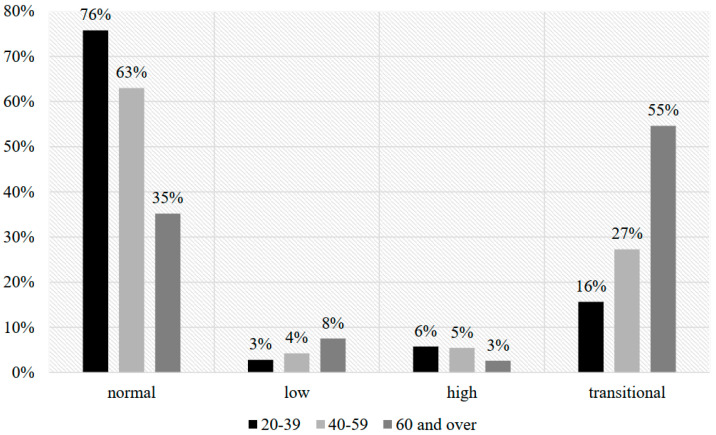
Bar chart for prevalence of HRV phenotypes by age group.

**Table 1 ijerph-21-01653-t001:** Upper and lower limits for heart rate and heart rate variability indices [15].

Indications	Aged 20–39	Aged 40–59	Aged 60 and Above
SDNN, ms	79–219	68–186	57–159
RMSSDs, ms	15–63	11–45	8–32
HR (bpm)	54–102	55–99	48–96

HR, bpm—heart rate, beats per minute. RMSSDs, ms—Root Mean Square of Successive Differences between normal RR intervals in a 24 H electrocardiogram recording, expressed in milliseconds (ms); SDNN, ms—standard deviation of all normal RR intervals in a 24 H electrocardiogram recording, expressed in milliseconds (ms).

**Table 2 ijerph-21-01653-t002:** Distribution of study participants by age groups and prevalence of CVD, DM, and CAN.

Name	Total Number	Aged 20–39			Aged 40–59			Aged 60 and Older		
		Total	Men	Women	Total	Men	Women	Total	Men	Women
Number of enrolled population	25.454	11.152	4790	6362	10.797	4699	6098	3505	1634	1871
Number of participants	5707	2486	1063	1423	2423	1056	1367	798	365	433
Prevalence of CVD per 10,000 population	2672.16	253.42	442.14	112.44	2558.71	3285.98	1997.071	10,551.38	10,684.93	10,438.8
Prevalence of DM per 10,000 population	527.42	48.27	65.85	35.14	548.91	653.41	468.18	1954.89	2328.77	1639.72
Prevalence of CAN per 10,000 population	1718.94	888.98	997.18	808.15	1939.74	2102.27	1814.19	3634.09	4027.4	3302.54

**Abbreviations:** CVD—cardiovascular diseases; DM—diabetes mellitus; CAN—cardiac autonomic neuropathy.

**Table 3 ijerph-21-01653-t003:** Comparison of HRV indicators with and without CAN.

No CAN				There Is CAN			
Indicators	SDNN	RMSSD	HR, bpm	Indicators	SDNN	RMSSD	HR, bpm
Total group							
Women (n = 2717)	144.9 ± 34.7	35.9 ± 14.8	71.7 ± 8.5	Women (n = 506)	85.3 ± 21.1	10.6 ± 4.4	86.6 ± 11.1
Men (n = 2009)	156.4 ± 39.5	38.4 ± 16.2	66.6 ± 8.4	Men (n = 475)	84.6 ± 22.6	10.7 ± 4.3	87.2 ± 10.8
*p*	<0.01	<0.01	<0.01	*p*	0.61	0.91	0.40
Aged 20–39							
Women (n = 1308)	155.5 ± 34.4	41.5 ± 14.6	73.2 ± 9.3	Women (n = 115)	91.6 ± 20.8	14.2 ± 5.6	90.8 ± 9.6
Men (n = 957)	172.2 ± 36.3	45.4 ± 14.9	65.8 ± 8.1	Men (n = 106)	84.6 ± 25.9	13.8 ± 2.7	88.8 ± 10.0
*p*	<0.01	<0.01	<0.01	*p*	0.03	0.53	0.12
Aged 40–59							
Women (n = 1119)	137.9 ± 32.7	30.8 ± 12.6	70.5 ± 7.2	Women (n = 248)	87.3 ± 18.7	10.6 ± 3.1	85.1 ± 10.4
Men (n = 834)	144.7 ± 37.1	32.3 ± 14.6	67.5 ± 8.6	Men (n = 222)	88.5 ± 21.2	10.7 ± 3.9	85.9 ± 11.1
*p*	<0.01	0.02	<0.01	*p*	0.50	0.62	0.39
Aged 60							
Women (n = 290)	124.1 ± 26.3	29.9 ± 14.1	69.5 ± 8.1	Women (n = 143)	76.8 ± 22.9	7.9 ± 2.97	85.7 ± 12.5
Men (n = 218)	131.8 ± 33.7	30.8 ± 14.5	66.4 ± 8.9	Men (n = 147)	78.6 ± 20.9	8.3 ± 4.28	87.8 ± 10.7
*p*	0.004	0.50	<0.01	*p*	0.47	0.36	0.12

**Abbreviations:** RMSSDs, ms—Root Mean Square of Successive Differences, expressed in milliseconds (ms), which represents the average squared differences between successive normal RR intervals from a 24 H ECG recording; SDNN, ms—standard deviation of normal-to-normal RR intervals, expressed in milliseconds (ms), which represents the average standard deviation of all normal RR intervals from a 24 H ECG recording; HR, bpm—heart rate, beats per minute.

## Data Availability

The data presented in this study are available on request from the corresponding author due to the protection of primary data.

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
