# Peer review of "Heart Rate Variability in Relation to Cardiovascular Autonomic Neuropathy Among Patients at an Urban Hospital in Kazakhstan"

_ijerph, 2024, doi:10.3390/ijerph21121653_

Round 1
Reviewer 1 Report
Comments and Suggestions for Authors
Heart Rate Variability in Relation to Cardiovascular Autonomic Neuropathy Among Patients at an Urban Hospital in Kazakhstan
General comment
In this study, SDNN and RMSSD recordings of 5707 participants were classified according to the criteria of a previous study (Umetani et al., 1998) and the result of the classification (phenotyping) was compared with the diagnosis of cardiovascular autonomic neuropathy (CAN). The authors concluded that HRV recordings are useful in a preventative measure for CAN. The reviewer considered that although the study was a worthwhile attempt, the manuscript needed significant improvement in its formatting.
Specific comments
Tables are formatted incorrectly and are unreadable.
The numerical results shown in Table 2 are incorrect.
Incorrect use of commas and periods in the numerical notation makes the results unintelligible.
Author Response
Dear Reviewer 1,
Thank you very much for your comments and recommendations! Thanks to your reviews, our manuscript has greatly improved. Here are our point-by-point responses to your comments.
REVIEWER 1
Comment Number |
Reviewers Comment |
1. |
General comment In this study, SDNN and RMSSD recordings of 5707 participants were classified according to the criteria of a previous study (Umetani et al., 1998) and the result of the classification (phenotyping) was compared with the diagnosis of cardiovascular autonomic neuropathy (CAN). The authors concluded that HRV recordings are useful in a preventative measure for CAN. The reviewer considered that although the study was a worthwhile attempt, the manuscript needed significant improvement in its formatting. Specific comments Tables are formatted incorrectly and are unreadable. The numerical results shown in Table 2 are incorrect. Incorrect use of commas and periods in the numerical notation makes the results unintelligible. |
Author’s Response |
|
The tables have been reformatted. In Table 2: two errors in numerical values have been corrected, and commas have been changed to periods. All the commas and periods in all the tables have been re-checked and been corrected. Moreover, the text in the manuscript was improved in its syntax, format, and made clearer and more comprehensive. |
Reviewer 2 Report
Comments and Suggestions for Authors
Although the article is well structured and lucid, there are a few grammatical issues and minor flaws that can be corrected. Such as
1. Inconsistent Use of Abbreviations**: The first mention of "HRV" should be defined as "heart rate variability (HRV)" before using the abbreviation.
2. The phrase "provide additional information, they do not necessarily offer better risk stratification [18]." could be made simpler by rephrasing it.
3. Some sentences are overly long and complex, which may hinder readability. For example, breaking down "This observation suggests a more specific impact of Dysglycemia and hyperglycemia on the onset and subsequent development of AN in the Kazakh population”.
4. The term "autonomic nervous system" appears multiple times in close succession; consider using "ANS" after the first mention to avoid redundancy.
5. The general typos in the text (such as use of double full stops in tables pg 7 of 16, use of comma between numbers pg 10 of 16)
Comments on the Quality of English LanguageMinor grammatical corrections and typos should be corrected.
Author Response
Dear Reviewer 2,
Thank you very much for your comments and recommendations! Thanks to your reviews, our manuscript has greatly improved. Here are our point-by-point responses to your comments.
REVIEWER 2
Comment Number |
Reviewers Comment |
1. |
Although the article is well structured and lucid, there are a few grammatical issues and minor flaws that can be corrected. Such as 1. Inconsistent Use of Abbreviations**: The first mention of "HRV" should be defined as "heart rate variability (HRV)" before using the abbreviation. 2. The phrase "provide additional information, they do not necessarily offer better risk stratification [18]." could be made simpler by rephrasing it. 3. Some sentences are overly long and complex, which may hinder readability. For example, breaking down "This observation suggests a more specific impact of Dysglycemia and hyperglycemia on the onset and subsequent development of AN in the Kazakh population”. 4. The term "autonomic nervous system" appears multiple times in close succession; consider using "ANS" after the first mention to avoid redundancy. 5. The general typos in the text (such as use of double full stops in tables pg 7 of 16, use of comma between numbers pg 10 of 16) |
Author’s Response |
|
1. The first mention of “HRV” was defined as “heart rate variability (HRV)”: Abstract: In clinical practice, heart rate variability (HRV) has not been considered an indicator for the preventive assessment of cardiovascular autonomic neuropathy (CAN). Introduction: The causal relationship between heart rate variability (HRV) and cardiovascular diseases (CVD) remains unclear despite the growing number of studies. Moreover, all the abbreviations were rechecked to achieve a consistent and coherent use of abbreviations in the text. 2. and 3. Were agree that some sentences had been too long and complex, which can make reading difficult. Thus, the complex and difficult to comprehend sentences and paragraphs have been rewritten to make our point clearer, more succinct, and easier to follow. 4. "Autonomic nervous system" was abbreviated into “ANS” after its first use in the text. 5. The general typos and mistakes have been corrected. The text became more clear and generally easier to understand. |
Reviewer 3 Report
Comments and Suggestions for Authors
This work describes two time-domain heart rate variability (HRV) indices in a large sample of patients from Kazakhstan, grouped by several criteria, including cardiac autonomic neuropathy (CAN) diagnosis and hypertension. Four groups of distinct characteristics, called phenotypes, are proposed and described. The work is original and relevant for readers of IJERPH. However, several issues should be addressed, as described below.
1. The study aim is not in agreement with the study design. Line 16 and line 63 say that the aim was to investigate changes as a marker for cardiovascular autonomic neuropathy. However, in the manuscript, there is no comparison against a gold standard that shows the diagnostic value of the supposed biomarkers based on HRV indices.
2. Methods section.
a) Describe all the study variables, including comorbidities.
b) Separate the ECG recording and HRV analysis on a different sub-section starting from line 84.
c) Lines 105 to 108. More information is needed regarding the ECG processing technique. What software was used? Indicate how it was validated. Please cite the Task Force 1996 reference for HRV recommendations (PMID: 8598068). Technical details such as the sampling frequency are needed. Please specify if only RR intervals from sinus rhythm were included in the analysis.
(d) Line 112. Add the definition of SDNN and RMSSD and the reference to the Task Force of 1996.
(e) Table 1. Add the appropriate reference for other reference values on the table header.
(f) Line 123. It says: "...HRV was at the average level". Which average? The average of the same five-minute segment, the average for the whole recording, or the average of the age group? Please be specific and clear about this classification. Also, please indicate how this criteria has been validated previously
(g) Lines 128 to 132. The statistical analysis needs a deep revision. First, indicate how the normal distribution was tested in all variables. If a variable had no normal distribution, non-parametric methods should be used to describe the results (median, percentiles 25 and 75) and to compare the groups. Second, indicate all the analyses performed, including the different comparisons. Third, the comparisons with more than two groups (for instance, by age) should be performed with ANOVA followed by posthoc adjustment of p-values (or the equivalent in non-parametric tests).
3. Results.
a) Table 2. What does the comma indicate? For instance, what does it mean to have a prevalence of 255,88? The prevalence should be expressed as a proportion or a percentage of the total sample.
b) Clarify what is the rationale for analysing the results only for subgroups of patients, for instance, patients without cardiac autonomic neuropathy (Table 3), patients with hypertension (Table 4),
c) Table 5. The description of the characteristics of phenotypic groups is entirely inefficient. It takes too much space and is hard to read. I suggest describing each phenotypic group narratively, which should take only one paragraph per phenotype. If the mentioned description appears after Table 5 and no additional information is reported on Table 5, delete the table.
4. Discussion.
a) Line 231. What does "decreased activity" mean? Please clarify.
b) Line 339 to 340. The sentence "Disproportionate distribution of subjects by sex and age was accounted for in the sample adjustment." What do you mean? Which sample adjustment?
5. Conclusions. Revise the first paragraph; there is no analysis against a gold standard to support the claim of the results as information for early diagnosis and treatment of autonomic neuropathy. This paragraph should be deleted from the conclusions since it is speculative, and such ideas could only be used in the Discussion section at best.
Comments on the Quality of English LanguageThe quality of English language is fine.
Author Response
Dear Reviewer 3,
Thank you very much for your comments and recommendations! Thanks to your reviews, our manuscript has greatly improved. Here are our point-by-point responses to your comments.
REVIEWER 3
Comment Number |
Reviewers Comment |
1. |
This work describes two time-domain heart rate variability (HRV) indices in a large sample of patients from Kazakhstan, grouped by several criteria, including cardiac autonomic neuropathy (CAN) diagnosis and hypertension. Four groups of distinct characteristics, called phenotypes, are proposed and described. The work is original and relevant for readers of IJERPH. However, several issues should be addressed, as described below. 1. The study aim is not in agreement with the study design. Line 16 and line 63 say that the aim was to investigate changes as a marker for cardiovascular autonomic neuropathy. However, in the manuscript, there is no comparison against a gold standard that shows the diagnostic value of the supposed biomarkers based on HRV indices. |
Author’s Response |
|
1. The aim of the study has been clearly identified in a rewritten manuscript. The abstract (lines 15-16) and introduction (63-64) now say that the aim of this study was to examine changes in heart rate variability (HRV) in a large randomly selected group of patients within the context of real clinical practice. |
|
2. |
Reviewers Comment |
2. Methods section. a) Describe all the study variables, including comorbidities. b) Separate the ECG recording and HRV analysis on a different sub-section starting from line 84. c) Lines 105 to 108. More information is needed regarding the ECG processing technique. What software was used? Indicate how it was validated. Please cite the Task Force 1996 reference for HRV recommendations (PMID: 8598068). Technical details such as the sampling frequency are needed. Please specify if only RR intervals from sinus rhythm were included in the analysis. (d) Line 112. Add the definition of SDNN and RMSSD and the reference to the Task Force of 1996. (e) Table 1. Add the appropriate reference for other reference values on the table header. (f) Line 123. It says: "...HRV was at the average level". Which average? The average of the same five-minute segment, the average for the whole recording, or the average of the age group? Please be specific and clear about this classification. Also, please indicate how this criteria has been validated previously (g) Lines 128 to 132. The statistical analysis needs a deep revision. First, indicate how the normal distribution was tested in all variables. If a variable had no normal distribution, non-parametric methods should be used to describe the results (median, percentiles 25 and 75) and to compare the groups. Second, indicate all the analyses performed, including the different comparisons. Third, the comparisons with more than two groups (for instance, by age) should be performed with ANOVA followed by posthoc adjustment of p-values (or the equivalent in non-parametric tests). |
|
Author’s Response |
|
a) In the study, the dependent variables included temporal heart rate variability (HRV) parameters: the average heart rate, the mean of the standard deviations of all normal-normal intervals (NN intervals) for 5-minute segments of a 24-hour ECG recording, and the square root of the average sum of the squares of the differences between adjacent NN intervals. Other dependent variables included gender (male or female), age groups ("20-39 years," "40-59 years," "60 years and older"), and the presence of diabetes (DM) and cardiac autonomic neuropathy (CAN). Additionally, the independent variable was the participants' status as public servants, based on patient assignment rules for the hospital. This also considered the prevalence of cardiovascular disease (CVD), diabetes (DM), and autonomic diabetic neuropathy (CAN) among the hospital's patient population. b) Now the method section has a section 2.2. 24-hour Holter Electrocardiogram Registration and HRV Analysis. c) Information regarding the ECG processing technique was added in detail in the section 2.2. 24-hour Holter Electrocardiogram Registration and HRV Analysis – lines 91-114. Technical specifications of the recorder include a QRS duration of 250 μs, a dynamic range of 12-14 mV, an analog bandwidth of 1.6 kHz, a lower filter frequency of 0.045 Hz, and a sampling rate of 4000-8000 Hz. Calibration for measurement accuracy of the recorder and software was performed the day before the study in November 2022. ECG recordings were processed both automatically and manually, adjusting any inappropriate computer labels. All 24-hour RR interval data from sinus rhythm, from one heartbeat to the next, were divided into 5-minute segments, resulting in 288 segments of 5-minute RR data. Heart rate variability (HRV) was assessed using time-domain analysis. d) The definitions of SDNN and RMSSD and the reference to the Task Force of 1996 were added in section 2.2. 24-hour Holter Electrocardiogram Registration and HRV Analysis – 93-103. e) The appropriate reference for other reference values on the table 1 header was added. f) Signs of the subclinical phase of CAN included the detection of below-average values for one of the indicators, either SDNN or RMSSD, along with a normal average heart rate – lines 127-134 (just after a table one). g) The normality of the distribution was assessed using the Shapiro-Wilk test. The analysis was performed with Mann–Whitney U test. Comparisons were made between two groups: males and females. In different age groups, comparisons were conducted separately rather than together, thus the test does not necessitate ANOVA. |
|
3. |
Reviewers Comment |
3. Results. a) Table 2. What does the comma indicate? For instance, what does it mean to have a prevalence of 255,88? The prevalence should be expressed as a proportion or a percentage of the total sample. b) Clarify what is the rationale for analysing the results only for subgroups of patients, for instance, patients without cardiac autonomic neuropathy (Table 3), patients with hypertension (Table 4), c) Table 5. The description of the characteristics of phenotypic groups is entirely inefficient. It takes too much space and is hard to read. I suggest describing each phenotypic group narratively, which should take only one paragraph per phenotype. If the mentioned description appears after Table 5 and no additional information is reported on Table 5, delete the table. |
|
Author’s Response |
|
a) The table 2 has been amended to address your questions – lines 158-159. b) After your comments changes have been made (two tables have been reformatted into one). The differences in the studied HRV indicators based on CAN are presented in Table 3 – lines 160-176. c) The table 5 has been removed. The narrative descriptions of the phenotypic groups correspond to the data in Tables 2 and 3. |
|
4. |
Reviewers Comment |
4. Discussion. a) Line 231. What does "decreased activity" mean? Please clarify. b) Line 339 to 340. The sentence "Disproportionate distribution of subjects by sex and age was accounted for in the sample adjustment." What do you mean? Which sample adjustment? |
|
Author’s Response |
|
a) The sentence has been clarified, – “…participants without CAN, predominantly represented in the "Normal HRV" phenotypic group, reflected the physiological autonomic activity of the cardiovascular system in both men and women: decreased sympathetic activity and parasympathetic dominance with age” – lines 222-225. b) The confusing sentence has been removed from the amended manuscript. The word “adjustment” meant the sample of study participants was selected in proportion to the population attached to the hospital. |
|
5. |
Reviewers Comment |
5. Conclusions. Revise the first paragraph; there is no analysis against a gold standard to support the claim of the results as information for early diagnosis and treatment of autonomic neuropathy. This paragraph should be deleted from the conclusions since it is speculative, and such ideas could only be used in the Discussion section at best. |
|
Author’s Response |
|
The conclusion has been rewritten – 328-337. |
Round 2
Reviewer 1 Report
Comments and Suggestions for Authors
General comment
The poor formatting and writing of this manuscript makes it difficult to read. Not only that, there are many inconsistencies in this paper. It needs thorough improvement for publication.
Specific comments
1. Umetani et al. estimated 95% CIs for each 10 years of age. For example, the CI for SDNN at age 20 is 93-257, at age 30 is 86-237, and at age 40 is 79-219. In Table 2 of this study, the CI for SDNN at 20-39 years of age is 79-219. How did you get the 95% CIs in Table 1 from Umetani's data?
2. The study by Umetani et al (1997) is the result of 24-hour ECGs from 260 participants. In general, more data are needed to estimate the 95% CI limit than to estimate the mean. Certainly, the study is of great historical importance, but the number of data was probably not sufficient to estimate the 95% CI limit. On the other hand, the present study is the result of 5707 participants and included 20 times more data than the study by Umetani et al. The reviewers considered that the results of this study should be used for the phenotyping threshold rather than the results of Umetani et al.
3. In Table 2, Aged 42-59 might bet Aged 40-59, Aged 60 might be Aged 60 or over.
4. The definition of Prevalence in the Method is incorrect (cases per 1,000).
5. L137 ‘The signs of the subclinical phase of the autonomic nervous system (ANS) were identified as the detection of below-average values of one of the indicators, ....’. However, average values were not shown in table 1.
6. Table 1 was entitled ‘Daily Variability of Heart Rate and Heartbeats within the Normal Range’. Does this table show “daily variability” ? Shouldn't it be “Upper and lower limits for heart rate and heart rate variability indices” .
7. There are no figures or tables to support the results of phenotyping that might be the most important finding of this study.
8. L416- “phenotyping based on HRV parameters identified AN in 17.19% of participants.” “17.19%” is not a result identified by the phenotyping.
9. Although the results of Table 4 and statistical tests are made for gender, the reviewer considered that gender differences were not major purpose of this study. Rather than the gender differences, the effect of CAN should be statistically tested.
10. The title of Table 4 is ‘Patients with Hypertension’. Despite this, all subjects (5707) are included in table 3.
11. “3.2 Phenotypic Analysis of Patients with CAN” This title is incorrect. This analysis includes people without CAN.
Author Response
Dear Reviewer 1,
Thank you very much for your comments and recommendations! Thanks to your reviews, our manuscript has greatly improved. Here are our point-by-point responses to your comments.
REVIEWER 1
Comment Number |
Reviewers Comment |
1. |
The poor formatting and writing of this manuscript make it difficult to read. Not only that, there are many inconsistencies in this paper. It needs thorough improvement for publication. Specific comments 1. Umetani et al. estimated 95% CIs for each 10 years of age. For example, the CI for SDNN at age 20 is 93-257, at age 30 is 86-237, and at age 40 is 79-219. In Table 2 of this study, the CI for SDNN at 20-39 years of age is 79-219. How did you get the 95% CIs in Table 1 from Umetani's data? |
Author’s Response |
|
The manuscript was revised to make it more concise, clear, and readable. The logical and stylistic inconsistencies were fixed.
1.The table reflects the heart rate and heart rate variability indicators (Umetani et al.), selected according to the predominant age of subjects within age groups. In the 20-39 age group, the proportion of subjects aged 30-39 was 64%, and for this reason, the SDNN values were indicated as "79–219," the RMSSD in milliseconds as "15-63," and the heart rate as 54-102. In the 40-59 age group, the proportion of subjects aged 50-59 was also predominant—52%. In the 60 and older group, the predominant age was 70 and older. |
|
2. |
Reviewers Comment |
The study by Umetani et al (1997) is the result of 24-hour ECGs from 260 participants. In general, more data are needed to estimate the 95% CI limit than to estimate the mean. Certainly, the study is of great historical importance, but the number of data was probably not sufficient to estimate the 95% CI limit. On the other hand, the present study is the result of 5707 participants and included 20 times more data than the study by Umetani et al. The reviewers considered that the results of this study should be used for the phenotyping threshold rather than the results of Umetani et al. |
|
Author’s Response |
|
The results of the study by Umetani et al. (1997) were applied in our research because they objectively reflect HRV indicators in healthy individuals and their fluctuations depending on sex and age. Their objectivity has been repeatedly confirmed in subsequent studies by other researchers. Given the above, we decided that the presence of similar changes in our study may indicate the objectivity of the method used to obtain the data. |
|
3. |
Reviewers Comment |
3. In Table 2, Aged 42-59 might bet Aged 40-59, Aged 60 might be Aged 60 or over. |
|
Author’s Response |
|
This is how we applied the results of the Umetani et al. (1997) study in forming the age groups of the research participants: the age range 42-59 years can be considered equivalent to 40-59 years, and the age of 60 years can be considered equivalent to 60 years and older. When selecting the indicators, we focused on the age that predominated within each age group. Your recommendations were taken into account when making changes to Table 1. |
|
4. |
Reviewers Comment |
4. The definition of Prevalence in the Method is incorrect (cases per 1,000). |
|
Author’s Response |
|
The definition was fixed to 100,000. |
|
5. |
Reviewers Comment |
5. L137 ‘The signs of the subclinical phase of the autonomic nervous system (ANS) were identified as the detection of below-average values of one of the indicators, ....’. However, average values were not shown in table 1. |
|
Author’s Response |
|
The line was changed to “The signs of the subclinical phase of the autonomic nervous system (ANS) were identified as the detection of below the lower…”. |
|
6. |
Reviewers Comment |
6. Table 1 was entitled ‘Daily Variability of Heart Rate and Heartbeats within the Normal Range’. Does this table show “daily variability”? Shouldn't it be “Upper and lower limits for heart rate and heart rate variability indices”. |
|
Author’s Response |
|
The table title was changed to “Upper and lower limits for heart rate and heart rate variability indices”. |
|
7. |
Reviewers Comment |
7. There are no figures or tables to support the results of phenotyping that might be the most important finding of this study. |
|
Author’s Response |
|
The figure one was added. |
|
8. |
Reviewers Comment |
8. L416- “phenotyping based on HRV parameters identified AN in 17.19% of participants.” “17.19%” is not a result identified by the phenotyping. |
|
Author’s Response |
|
The sentence was removed. The line “CAN was identified in 17.19% of the study participants, with a ratio of the subclinical to clinical phase of 1:0.24” was added to the abstract. |
|
9. |
Reviewers Comment |
9. Although the results of Table 4 and statistical tests are made for gender, the reviewer considered that gender differences were not major purpose of this study. Rather than the gender differences, the effect of CAN should be statistically tested. |
|
Author’s Response |
|
We apologize for the mistake, but Table 4 shouldn’t have been included in the revised version of the manuscript. Secondly, in Table 3, the statistical tests show both gender differences (represented by vertical columns) and differences between the cohorts of participants with and without CAN. The changes identified through the statistical tests allowed us to form the differences between the phenotypic groups (Section 3.2). |
|
10. |
Reviewers Comment |
10. The title of Table 4 is ‘Patients with Hypertension’. Despite this, all subjects (5707) are included in table 3. |
|
Author’s Response |
|
Table 4 is not included in the latest version of the manuscript, in accordance with the Reviewer’s recommendations during the previous manuscript revision. |
|
11. |
Reviewers Comment |
11. “3.2 Phenotypic Analysis of Patients with CAN” This title is incorrect. This analysis includes people without CAN. |
|
Author’s Response |
|
Section 3.2 was changed to “Phenotypic Analysis of Study Participants”. |
Reviewer 3 Report
Comments and Suggestions for Authors
The changes made by the authors improved significantly the manuscript.
I consider that the updated manuscript is now suitable for publication. I agree with the publication of the manuscript.
Comments on the Quality of English LanguageNone
Author Response
We thank you for your review!